# Stereoselective one-pot synthesis of polypropionates

Guo-Ming Ho[1], Medel Manuel L. Zulueta [1] & Shang-Cheng Hung[1]

Polypropionates—motifs with alternating methyl and hydroxy groups—are important segments of many natural products possessing high bioactivity and therapeutic value. Synthetic access to these structures remains an area of intensive interest, focusing on the establishment of the contiguous stereocentres and a desire for operational simplicity. Here we report an efficient strategy for the stereoselective assembly of polypropionates with three or four stereocentres through a three-step relay process that include Diels–Alder reaction, silylenol ether hydrolysis and Baeyer–Villiger oxidation. The stereochemistry and functionality of the resulting polypropionates depend on the substitution pattern of the diene and dienophile substrates of the Diels–Alder cycloaddition. More importantly, the relay sequence is effectively performed in one pot, and the product could potentially undergo the same sequence for further elaboration. Finally, the C1–C9 segment of the macrolide etnangien is constructed with four of the six stereogenic centres established using the relay sequence.

[1] Genomics Research Center, Academia Sinica, 128, Section 2, Academia Road, Taipei 115, Taiwan. Correspondence and requests for materials should be addressed to S.-C.H. (email: schung@gate.sinica.edu.tw)

Polyketides are a large class of structurally diverse secondary metabolites isolated from fungi, bacteria and sponges[1]. They exhibit a wide range of biological activities, including antibiotic, antitumour, antifungal, cholesterol-lowering, antiparasitic and immunosuppressive properties[1, 2]. In particular, several polyketide-containing natural products, such as the antibiotics erythromycin[3] and monensin A[4] and the antitumour discodermolide[5] (Fig. 1), have displayed clinical significance. A common feature of natural products with polyketide structures are polypropionate segments characterized by alternating methyl and hydroxy groups[6]. The multiple stereogenic centres displayed in these segments provide for large numbers of possible stereochemical permutations. A stereotetrad motif (structure with four contiguous stereogenic centres), for example, could have 16 possible diastereomers. Consequently, the preparation of polypropionates has been a longstanding interest in the synthesis community[6–10].

Numerous synthetic approaches have been developed to address the architectural complexity of polypropionates. Forging a single C–C bond into adjacent methyl and hydroxy-bearing stereogenic centres could be accomplished by established aldol reactions[11], but crotylation[12–14], allenylation[15], epoxide opening[6], [2 + 2]-cycloaddition[16] and many other methods[6] are available. Although the application of these methods in iterative protocols allowed for more complex polypropionate subunits, additional step-consuming transformations are required for each subsequent C–C bond extensions. In addition, the stereochemical control of the newly created stereocentres over the course of the propionate chain elongation is often complicated. Attractive alternatives, especially considering atom and step economy, are multicomponent domino (or cascade) reactions[17–19] that could create more than one C–C bond, and hence, multiple stereocentres in a single-pot operation. Nevertheless, only particular sets of stereoisomers could be achieved in these

**Fig. 1** Polypropionate-containing natural products. The major polypropionate segments are marked by dashed polygons

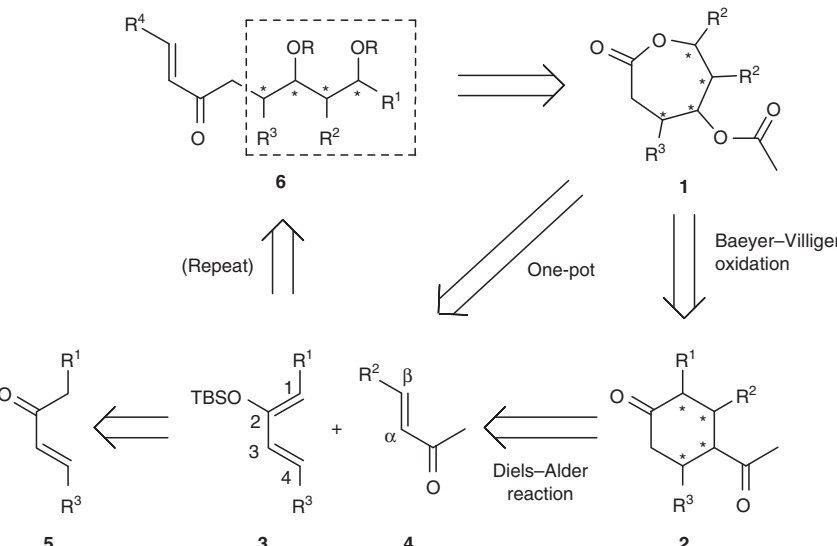

**Fig. 2** Our retrosynthetic analysis for the synthesis of polypropionates. The asterisks represent possible stereogenic centres. TBS *tert*-butyldimethylsilyl

**a**

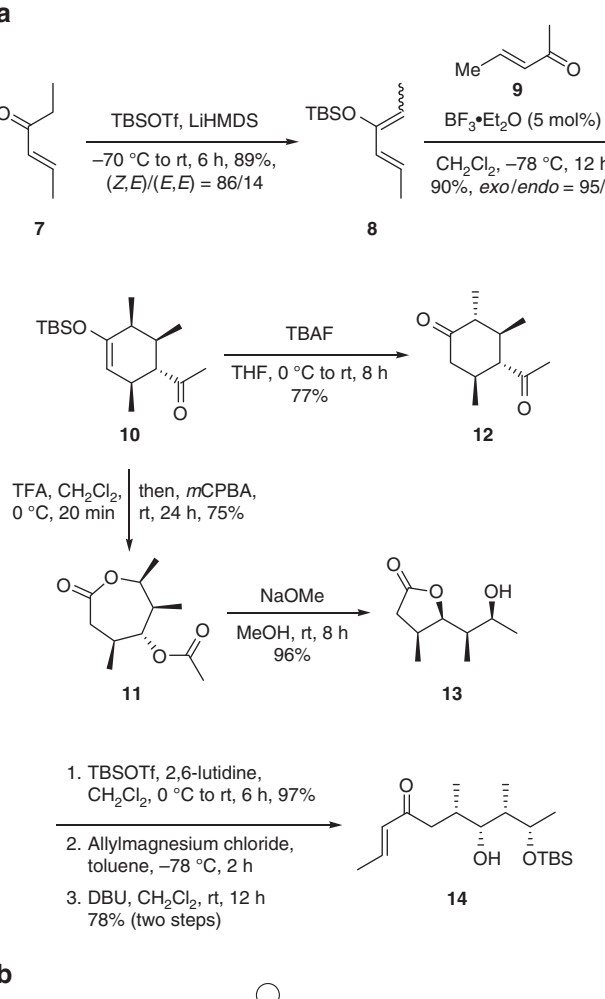

**b**

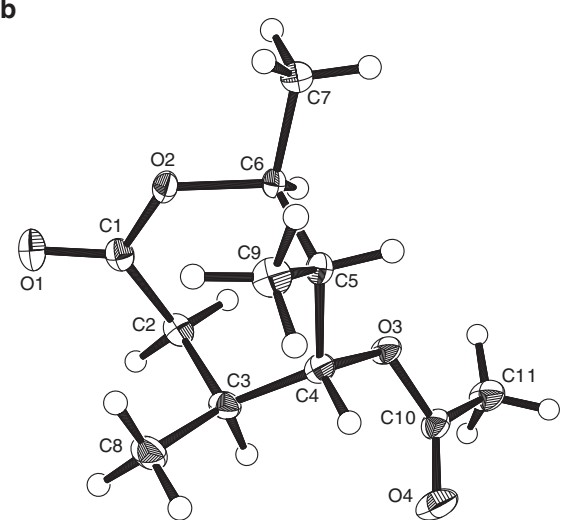

**Fig. 3** Stepwise preparation of polypropionate **14**. **a** Synthetic route. **b** X-ray crystal structure of **11**. DBU 1,8-diazabicyclo[5.4.0]undec-7-ene; LiHMDS lithium hexamethyldisilazide; *m*CPBA, *meta*-chloroperoxybenzoic acid; TBAF, tetrabutylammonium fluoride; Tf, triflyl; TFA, trifluoroacetic acid; THF, tetrahydrofuran

procedures. Thus, further development of convenient entries to several possible stereotriads, stereotetrads or stereopentads, as well as their related structures that exploit operationally simple strategies in library design and drug discovery of polypropionate-containing molecules are still highly desirable.

Motivated by the intensive interest towards the expedient synthesis of polypropionates, we herein devise an alternative stereoselective strategy from simple and readily available precursors. This approach involves sequential Diels–Alder cycloaddition of silylenol ethers and enones acting as the respective dienes and dienophiles, silylenol ether hydrolysis to afford a cyclic ketone and Baeyer–Villiger oxidation to enable double oxygen insertions with stereoretention at key positions. Ultimately, the relay sequence provides a suitable access into polypropionates with three or four contiguous stereocentres both in stepwise and one-pot manner. Using this procedure, the synthesis of the C1–C9 segment of the macrolide natural product etnangien is also successfully demonstrated.

## Results

**Synthetic strategy**. Our synthetic design follows the formation of the seven-membered lactone **1**, harbouring three to four stereocentres (Fig. 2). The embedded oxygen functionalities at C4 and C6 positions of lactone **1** can be traced back to cyclohexanone **2** by Baeyer–Villiger oxidation. Analysis of compound **2** shows possible access from a pivotal Diels–Alder cycloaddition of silyloxydiene **3** and dienophile **4**. Previous studies have demonstrated the strong effect of the terminal substitution of dienes and dienophiles in the *exo*- and *endo*-stereochemical outcome of the Diels–Alder reaction[20, 21]. Thus, we anticipate a favourable control of stereoselectivity during the formation of the cycloadduct. Finally, the required dienes, and in certain cases, the dienophiles could be afforded from the commercially or readily available enone **5**. This synthetic approach benefits from the inherent flexibility of the Diels–Alder reaction and the ready availability of various dienes and dienophiles, with the potential option to grow the polypropionate chains having the desired diastereomeric relationships. In addition, the transformations of the diene **3** and dienophile **4** towards the lactone **1**, that is, the Diels–Alder/Baeyer–Villiger relay process, could be performed using compatible reagents to permit not only stepwise handling but also one-pot operation. Furthermore, with synthetically versatile groups at the terminal positions of the corresponding open form of the lactone **1**, it is, in principle, possible to reiterate the process through compound **6** carrying the familiar enone structure.

**Evaluation of the Diels–Alder/Baeyer–Villiger relay process**. To examine the proposed relay concept, we first applied the designed procedure to the prototypical enones **7** and **9** in a stepwise manner (Fig. 3a). Deprotonation of enone **7** with lithium hexamethyldisilazide followed by silyl trapping supplied the silylenol ether **8** in 89% yield ((Z,E)/(E,E) = 86/14). Alternatively, use of triethylamine as base to permit the same reaction led to lower yield (82%) and diastereoselectivity for the desired (Z,E)-isomer (85/15). The cycloaddition of diene **8** and dienophile **9** is expected to produce the *exo*-adduct based on the methyl substitution at C1 of **8** and Cβ of **9**[21] (for the carbon designations of dienes and dienophiles used in this paper, refer to Fig. 2). Thus, the boron trifluoride-catalysed Diels–Alder reaction effectively provided the product **10** in excellent yield and *exo*-selectivity. Other Lewis acid catalysts, such as copper(II) triflate, scandium(III) triflate, trimethylsilyl triflate, dimethylaluminum chloride and tin tetrachloride, also gave similar *exo*-preference, but only in moderate-to-good yields (Supplementary Table 1). Notably, the supposed Diels–Alder cycloadduct between the (E,E)-isomer of **8** and the enone **9** was not observed in these cases. The desilylation of **10** could be achieved by tetrabutylammonium fluoride (TBAF), but acid hydrolysis offers better compatibility with Baeyer–Villiger oxidation in light

**Table 1 One pot operation for the tandem Diels–Alder and Baeyer–Villiger transformations**

| Entry | Diene | Enone | Lewis acid | Product | Yield (%)[a] |
|---|---|---|---|---|---|
| 1 | **8** ($R^1 = R^2 = Me$) | **9** ($R^3 = Me$) | $BF_3 \cdot Et_2O$ | **11** (type a) | 63 |
| 2 | **8** | **15** ($R^3 = H$) | $Cu(OTf)_2$ | **16** (type b) | 53 |
| 3 | **17** ($R^1 = (CH_2)_2OBn$, $R^2 = Me$) | **15** | $Cu(OTf)_2$ | **18** (type b) | 43 |
| 4 | **17** | **9** | $BF_3 \cdot Et_2O$ | **19** (type a) | 64 |
| 5 | **20** ($R^1 = CH_2Ph$, $R^2 = Me$) | **9** | $BF_3 \cdot Et_2O$ | **21** (type a) | 64 |
| 6 | **22** ($R^1 = Ph$, $R^2 = Me$) | **9** | $BF_3 \cdot Et_2O$ | **23** (type a) | 62 |
| 7 | **24** ($R^1 = CH_2iPr$, $R^2 = Me$) | **9** | $BF_3 \cdot Et_2O$ | **25** (type a) | 64 |
| 8 | **26** ($R^1 = CH_2iBu$, $R^2 = Me$) | **9** | $BF_3 \cdot Et_2O$ | **27** (type a) | 60 |
| 9 | **28** ($R^1 = Me$, $R^2 = H$) | **9** | $BF_3 \cdot Et_2O$ | **29** (type a) | 62 |
| 10 | **30** ($R^1 = (CH_2)_2OBn$, $R^2 = H$) | **9** | $BF_3 \cdot Et_2O$ | **31** (type a) | 61 |
| 11 | **28** | **32** ($R^3 = Ph$) | $BF_3 \cdot Et_2O$ | **33** (type a)/**34** (γ-lactone) | 45/18 |
| 12 | **30** | **32** | $BF_3 \cdot Et_2O$ | **35** (type a)/**36** (γ-lactone) | 42/14 |

[a]Based on product isolated after chromatographic separation

of our desire for a one-pot operation. Consequently, treatment with trifluoroacetic acid (TFA) for 20 min delivered the corresponding dione, which was directly treated with *meta*-chloroperoxybenzoic acid (*m*CPBA) for the double oxygen insertion to afford the expected ε-lactone **11** carrying a poly-propionate *syn,syn,syn*-stereotetrad in 75% yield (two steps, one pot). The structure of **11** was unambiguously confirmed by X-ray crystallographic analysis (Fig. 3b, Supplementary Table 2, Supplementary Data 1) and further characterized by nuclear magnetic resonance (NMR) spectroscopy (Supplementary Methods). Curiously, the exposure of compound **10** to TBAF led to a complete epimerization at the α-carbon leading to compound **12** (77%) as a single diastereomer, providing a possible point of access to another stereotetrad structure.

Treatment of the lactone **11** with catalytic sodium methoxide in methanol resulted in deacetylation and transesterification, supplying the γ-lactone **13** (96%). This cyclic system should offer more stability than the initial seven-membered ring structure and could serve as key intermediate with various potential for further manipulation. With the intent to explore the possibility of reiterating the tandem Diels–Alder/Baeyer–Villiger reaction process, we examined the elaboration of compound **13** to further generate an advanced enone structure. The operation started from the silyl group protection of the free hydroxy group (97%), followed by nucleophilic addition to the carbonyl using allylmagnesium chloride. Subsequent base-promoted double bond isomerization ultimately afforded the desired enone **14** (78% in two steps). This and similar compounds offer prospective applications for the assembly of longer and more complex polypropionate chains.

After successfully demonstrating stereotetrad formation via Diels–Alder cycloaddition and Baeyer–Villiger oxidation in stepwise manner, we then explored the feasibility of our

envisioned one-pot operation (Table 1). Pleasingly, the Diels–Alder *exo*-adduct, formed in situ from compounds **8** and **9** by the catalytic assistance of boron trifluoride, was smoothly converted to the desired ε-lactone **11** in 63% yield upon successive treatment with TFA and *m*CPBA in the same vessel (entry 1). For cycloaddition with enone **15**, we found that copper (II) triflate provided the best product yield and selectivity for the expected *endo*-adduct[20] (Supplementary Table 1). Thus, the sequential Diels–Alder cycloaddition of **8** and **15**, acid hydrolysis and Baeyer–Villiger oxidation supplied the ε-lactone **16** in 53% yield for the one-pot process (entry 2). Further application of the one-pot procedure to different diene and enone combinations all provided satisfactory results (entry 3–12). Curiously, when the phenyl-substituted **32** was employed as starting enone (entries 11 and 12), minor amounts of the separable corresponding γ-lactone (**34** and **36**), apparently resulting from acid-induced translacto-nization[22], was acquired along with the expected ε-lactone (**33** and **35**). The structures of the duly formed lactones were confirmed by NMR spectroscopy as well as X-ray crystallographic analysis for compounds **16**, **23**, **29** and **34** (Supplementary Fig. 1, Supplementary Tables 3–6, Supplementary Data 2–5). Treatment of all the lactone products with sodium methoxide in methanol resulted into the corresponding γ-lactone akin to compound **13** with variously substituted stereogenic centres (Supplementary Table 7).

**Preparation of the C1–C9 segment of etnangien**. To demon-strate the utility of the tandem one-pot Diels–Alder/Baeyer–Villiger relay process for the assembly of polypropionates leading to more complex molecular architectures, we targeted the stereohexad structure present at the C1–C9 subunit of the macrolide etnangien (Fig. 1). Etnangien, an antibiotic isolated

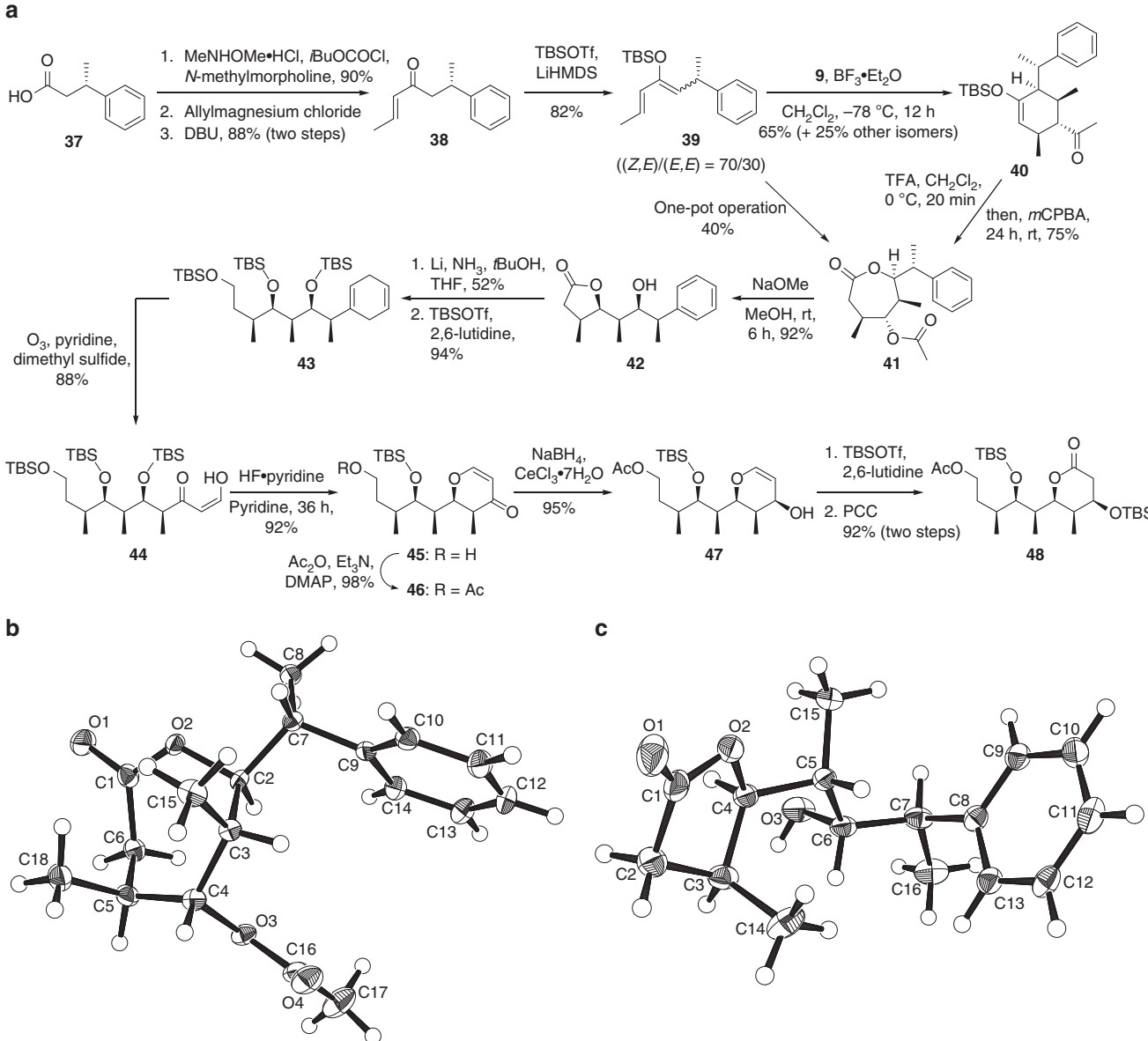

**Fig. 4** Synthesis of the C1–C9 fragment of etnangien. **a** Synthetic route. **b** X-ray crystal structure of **41**. **c** X-ray crystal structure of **42**. Ac, acetyl; DMAP, N,N-dimethyl-4-aminopyridine; PCC, pyridinium chlorochromate

from the myxobacterium *Sorangium cellulosum*, is a potent inhibitor of the RNA polymerase of Gram-positive bacteria[23]. Our synthesis began with the Weinreb amidation (85%) of the commercially available (S)-3-phenylbutanoic acid (**37**) (Fig. 4a). Nucleophilic allyl addition to the amide, followed by double-bond isomerization established the familiar enone structure **38**. Silylation of this enone fashioned the desired 1,4-disubstituted diene **39** in 82% yield ((Z,E)/(E,E) = 70/30).

With required diene **39** in hand, the formation of the target ε-lactone through the crucial relay process was first performed in a stepwise fashion. Thus, the Diels–Alder cycloaddition of the diene **39** and enone **9** employing the boron trifluoride-catalysed condition produced two *exo*- and two *endo*-cycloadduct diastereoisomers (*exo/endo* = 84/16) of which the desired silylenol ether and exo-adduct **40** was obtained as the major diastereomer in 65% yield; the other diastereomers were collectively afforded in 25% yield (see the Supplementary Methods for details). A preliminary X-ray analysis of the racemic version of **40** supported the structure of this major diastereomer (Supplementary Fig. 1,

Supplementary Table 8, Supplementary Data 6). The facial selectivity of the Diels–Alder reaction towards **40** is perceived as a result of the asymmetric induction of the stereocentre attached to C1 of the diene (Supplementary Fig. 2). Having the desired stereochemistry and the necessary carbon atoms of the target subunit, the *exo*-adduct **40** was subjected to sequential hydrolysis and Baeyer–Villiger oxidation utilizing TFA and *m*CPBA, respectively, generating the required ε-lactone **41** in 75% yield as a single isomer. X-ray crystallography confirmed the structure of **41** with the prescribed relative configuration of the contiguous stereocentres on the seven-membered lactone ring (Fig. 4b, Supplementary Table 9, Supplementary Data 7). After the successful stepwise attempt, we next pursued the transformation in one pot. To our satisfaction, the three-step sequence also proceeded smoothly and provided the expected lactone **41** in 40% overall yield. Treatment of **41** with sodium methoxide resulted to deacetylation and subsequent conversion into the γ-lactone **42** (96%). Again, an X-ray analysis supported the structure and

relative configuration of the stereopentad motif of the crystalline **42** (Fig. 4c, Supplementary Table 10, Supplementary Data 8).

With implicitly positioned terminal groups, **42** was further modified to form the more elaborated C1–C9 fragment of etnangien, particularly using a Birch reduction/ozonolytic cleavage sequence[24]. Thus, exposure of the alcohol **42** to lithium metal in the presence of ammonia[25] reduced the lactone into a terminal alcohol and the phenyl group into a 1,4-cyclohexadiene structure. A side product of this reaction, with the phenyl group remaining unscathed, was further converted to **42** using the same Birch reduction in moderate yield (Supplementary Methods). Silylation of the alcohol groups led to compound **43**. Ozonolysis transformed the 1,4-cyclohexadiene portion of **43** into the keto/*cis*-enol pair as verified by NMR spectroscopy of **44** with likely stabilization by hydrogen bonding. Treatment of **44** with hydrogen fluoride in pyridine provided for partial desilylation and subsequent enol ether formation via a lactol intermediate. Acetylation of the so-formed **45** followed by Luche reduction[26] established the alcohol **47** as a single diastereosomer in excellent yield. Final silylation and oxidation provided the desired C1–C9 fragment of etnangien containing six stereocentres (**48**). This fragment possesses the synthetic versatility that would enable further elaboration towards etnangien and other more advanced molecules.

## Discussion

We have developed an efficient approach to stepwise and one-pot stereoselective synthesis of polypropionate structures through a three-step Diels–Alder cycloaddition/Baeyer–Villiger oxidation relay process. The highly substrate-controlled Diels–Alder reaction, facilitated by the Lewis acid catalysts boron trifluoride and copper(II) triflate, enabled the establishment of the necessary cyclohexene ring featuring the silylenol ether and three-to-four stereogenic centres. Following an acid hydrolysis to provide the aptly positioned dicarbonyl groups, Baeyer–Villiger oxidation with *m*CPBA formed the lactone and ester functionalities. Using this procedure, a range of polypropionate stereoisomers with high stereochemical flexibility and substantial skeletal diversity was accessed. The amenability of our strategy for a one-pot operation offers a rapid and convenient method to the many possible polypropionate motifs as demonstrated herein. Importantly, the resulting products not only feature the polyketide subunits, but also render functionalities that could, in principle, be further transformed for use in subsequent C–C bond forming events or other elaborations to more complex structures. Further extension of this methodology to provide more comprehensive access to other biologically significant polypropionate motifs is ongoing and will be reported in due course.

## Methods

**One-pot preparation of compound 41.** Freshly distilled ketone **9** (0.24 mL, 2.38 mmol) in $CH_2Cl_2$ (2 mL) was added by syringe to a solution of $BF_3 \cdot OEt_2$ (12 μL, 99 μmol) in $CH_2Cl_2$ (2 mL) at −78 °C under argon atmosphere. After stirring at −78 °C for 10 min, diene **9** ((*Z,E*)/(*E,E*) = 70/30, 0.600 g, 1.99 mmol) in $CH_2Cl_2$ (2.0 mL) was added dropwise, and the reaction was allowed to stir for 12 h at the same temperature. TFA (0.18 mL, 2.38 mmol) was next added in one portion, and the reaction was allowed to warm to 0 °C. After 20 min at 0 °C, the resulting mixture was poured into a solution of *m*CPBA (77%, 3.56 g, 15.9 mmol) in $CH_2Cl_2$ (20 mL) at 0 °C. The reaction was warmed to ambient temperature and stirred for 24 h. The reaction mixture was slowly quenched with satd. $Na_2S_2O_{3(aq)}$ at 0 °C, and the crude organic portion was extracted with $CH_2Cl_2$. The combined organic layer was washed with satd. $NaHCO_{3(aq)}$ and brine, dried over $MgSO_4$, filtered and concentrated under reduced pressure. Purification via flash column chromatography (*n*-hexane/ethyl acetate = 4/1) provided the ε-lactone **41** (0.169 g, 40%) as a colourless solid. $[\alpha]^{23}_D = -7.28$ (*c* 10, $CHCl_3$); m.p. 130–131 °C (recrystallized from *n*-hexane/$CHCl_3$); Infrared Spectrum (thin film): ν 2973, 1734, 1637, 1454, 1239, 1183, 1087, 1020, 767, 703 cm$^{-1}$; $^1$H NMR (600 MHz, $CDCl_3$): δ 7.28 (t, *J* = 7.5 Hz, 2H; Ph-H), 7.21 (t, *J* = 7.2 Hz, 1H; Ph-H), 7.12 (d, *J* = 7.8 Hz, 2H; Ph-H), 4.70 (d, *J* = 10.6 Hz, 1H; ε-methine-H), 4.67 (s, 1H; γ-methine-H), 3.28 (dd, *J* = 14.0, 6.7 Hz, 1H; α-methylene-

H), 2.99 (dq, *J* = 13.4, 6.7 Hz, 1H; benzylic H), 2.44 (dd, *J* = 14.0, 5.2 Hz, 1H; α-methylene-H), 2.10 (br, 1H; β-methine-H), 1.90 (s, 3H, $CO_2CH_3$), 1.56–1.54 (m, 1H; δ-methine-H), 1.38 (d, *J* = 6.7 Hz, 3H; $CH_3$), 1.14 (d, *J* = 7.7 Hz, 3H; $CH_3$), 1.01 (d, *J* = 7.6 Hz, 3H; $CH_3$); $^{13}$C NMR (150 MHz, $CDCl_3$): δ 173.6 (C), 168.9 (C), 143.1 (C), 128.9 (CH × 2), 127.2 (CH × 2), 127.2 (CH), 80.9 (CH), 77.7 (CH), 41.9 (CH), 37.4 (CH), 35.1 (CH$_2$), 32.6 (CH), 21.1 (CH$_3$), 20.2 (CH$_3$), 16.9 (CH$_3$), 10.1 (CH$_3$); High Resolution Mass Spectrum (Electrospray Ionisation): *m/z* calcd for $C_{18}H_{24}O_4Na$ ([M + Na]$^+$): 327.1572, found: 327.1580. An X-ray analysis of the crystal supported the structure of **41** and the relative configuration of its contiguous chiral centres ((Fig. 4b, CCDC 1556026).

The complete experimental details and compound characterization data can be found in the Supplementary Methods. For the NMR spectra of the compounds in this article, see Supplementary Figs. 3–102.

**Data availability**. The X-ray crystallographic coordinates for compounds **11**, **16**, **23**, **29**, **34**, **40**, **41** and **42** in this study have been deposited at the Cambridge Crystallographic Data Centre (CCDC), under deposition numbers CCDC 1511846, CCDC 1511847, CCDC 1511848, CCDC 1511849, CCDC 1511880, CCDC 1511850, CCDC 1556026 and CCDC 1556034, respectively. These data can be obtained free of charge from the CCDC via www.ccdc.cam.ac.uk/data_request/cif. All data that support the findings of this study are available from the corresponding author upon reasonable request.

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

## Acknowledgements

This work was supported by the Ministry of Science and Technology (MOST 104-0210-01-09-02, MOST 104-2628-M-001-001, MOST 105-0210-01-13-01, MOST 105-2745-M-001-003-ASP) and Academia Sinica.

## Author contributions

S.-C.H. conceived the project and supervised staffs. G.-M.H. performed the synthetic experiments. M.M.L.Z. and G.-M.H. wrote the manuscript. All authors discussed the results and commented on the manuscript.

## Additional information

**Competing interests:** The authors declare no competing financial interests.

