## [Peer Review File · Nature Communications]

Reviewers' comments:

Reviewer #1 (Remarks to the Author):

Hung and coworkers describe a creative approach for the synthesis of fragments of polypropionates. The authors target a very important problem in natural product synthesis and have made remarkable progress toward it. The one-pot Diels–Alder reaction, silylenol ether hydrolysis, and Baeyer–Villiger oxidation is highly efficient and stereoselective. A series of polypropionate motifs have been synthesized through this approach, which proves that the methodology is robust. The weakness of this piece of work is the synthetic application, the synthesis of the C4–C9 fragment of etnangien. This model system is too simple to be useful in a real total synthesis program of etnangien. For example, the phenyl group should be replaced with a furyl or PMP group at least, which could be potentially oxidized to give a carboxylic acid as a reasonable handle. The C3-OH should be incorporated as well, which is important in a fully elaborated intermediate (with three methyl plus three hydroxyl groups). This manuscript could be accepted for publication in Nature Communications if a fully elaborated fragment of etnangien is constructed by using the methodology reported.

Reviewer #2 (Remarks to the Author):

This manuscript details a sequential one-pot synthetic strategy to the stereoselective formation of some polypropionates, using a Diels-Alder/Baeyer-Villiger approach. This is a solid study which has been competently carried out, and the conclusions are appropriately supported by the presented data.

The crystallographic study consists of structure reports of several of the key products and intermediates (7 structures), including the Etnangien fragment 42 and precursor 40. All of the structures are fairly routine collections and the structure models have been suitably refined; although the data quality for some (compounds 16 and 42, due to some collection issues and twinning, respectively) is not quite as good as the others. Nonetheless, the models provided are appropriately used to support the conclusions of the paper, which is well within the capacity of these structural models.

With a couple of minor corrections below, I believe this report will be appropriate for publication in Nature Communications.

Requested corrections:

- The experimental section is missing a discussion of the X-ray data collection and refinement: a standard discussion of the X-ray instrumentation and particular refinement strategies should be added, particularly describing the handling of the racemic twinning in compound 42
- It is unfortunate that the authors chose to use racemic 3-phenylbutanoic acid as the starting material for the synthesis of the Etnangien fragment; as the absolute configuration of the natural product itself is known, using the commercially-available enantiomerically pure starting material might have made for a stronger case study. Although I don't see this as a big limitation, in my view it is a little misleading in the supplementary information to represent intermediates S14, 38 and 39 with absolute configuration shown - this should be amended.

Reviewer #3 (Remarks to the Author):

This paper describes a diastereoselective sequential process for the synthesis of partial structures of polyketides. The process involves a Lewis acid-catalyzed Diels-Alder reaction between silyl enol

ethers and enones as the initial step. After Baeyer-Villiger oxidation in one-pot, target compounds are obtained. The process is applied to the synthesis of a partial structure of a natural product, etnangien.

This paper is not suitable for Nature Communications due to the following reasons.

1. No conceptual news: The Lewis acid-catalyzed exo-selective Diels-Alder reaction between silyl enol ethers and enones is well-known. The Baeyer-Villiger oxidation is known to proceed with stereo-retention. Therefore, the process described in this paper is simply a combination and logical extension of two fundamental, text-book level reactions.
2. No enantio-control: The process depends simple on relative stereochemical control, and absolute chirality control is not achieved. In the application to etnangien, racemic starting material 37 was used. Possible reason for the facial selectivity (diastereoselectivity) of the diene side is not described.
3. Exaggerated introduction: In the introductory section, the authors describe that repeating the process from product 6 (or 14) will provide more elaborated structures. However, this is not achieved. Although the authors imply that previous methods for polyketide synthesis are step-consuming and not of atom-economy, the reported process also required multiple steps with using not-atom economically friendly reagents. The authors mention that the desired stereoisomers can be synthesized at will, which is again not achieved in the paper.

Response to Referees

Reviewer 1

1. The weakness of this piece of work is the synthetic application, the synthesis of the C4–C9 fragment of etnangien. This model system is too simple to be useful in a real total synthesis program of etnangien. For example, the phenyl group should be replaced with a furyl or PMP group at least, which could be potentially oxidized to give a carboxylic acid as a reasonable handle. The C3-OH should be incorporated as well, which is important in a fully elaborated intermediate (with three methyl plus three hydroxyl groups). This manuscript could be accepted for publication in Nature Communications if a fully elaborated fragment of etnangien is constructed by using the methodology reported.

Answer: As stated in our original manuscript, the phenyl group of compound **42** could be transformed into a supposed 1,3-dicarbonyl functionality by a Birch reduction/ozonolytic cleavage sequence. This transformation sequence has been carried out and is now included in our revised manuscript. In this revision, we also performed the synthetic sequence using the enantiomerically pure (*S*)-3-phenylbutanoic acid instead of the racemic version to afford the correct absolute configuration of the etnangien fragment. Together with Luche reduction to establish the C3-OH functionality in proper stereochemistry and pyridinium chlorochromate oxidation to afford the lactone system, the end result is a C1–C9 fragment of etnangien (compound **48**) with six contiguous stereocentres. Compound **48** was also fitted with functional groups that could be useful in further elaborations toward etnangien.

Reviewer 2

1. The experimental section is missing a discussion of the X-ray data collection and refinement: a standard discussion of the X-ray instrumentation and particular refinement

strategies should be added, particularly describing the handling of the racemic twinning in compound **42**.

Answer: We have now expanded the General Methods section in the Supplementary Information to include X-ray data collection and refinement procedures.

2. It is unfortunate that the authors chose to use racemic 3-phenylbutanoic acid as the starting material for the synthesis of the Etnangien fragment; as the absolute configuration of the natural product itself is known, using the commercially-available enantiomerically pure starting material might have made for a stronger case study. Although I don't see this as a big limitation, in my view it is a little misleading in the supplementary information to represent intermediates S14, 38 and 39 with absolute configuration shown - this should be amended.

Answer: To match the absolute configuration of the etnangien fragment, we repeated the synthetic procedure using the enantiomerically pure (*S*)-3-phenylbutanoic acid through our Diels–Alder/Baeyer–Villiger relay process to afford the proper enantiomer **42**. Moreover, we further elaborated compound **42** and acquired the C1–C9 fragment of etnangien (**48**) possessing the appropriate stereochemistry and terminal handles amenable for future transformations. The details about this new synthesis are now included in our revised manuscript. The intermediates/products of this synthetic sequence now complement the structure representations provided in Fig. 4 and the Supplementary Information.

Reviewer 3

1. No conceptual news: The Lewis acid-catalyzed exo-selective Diels-Alder reaction between silyl enol ethers and enones is well-known. The Baeyer-Villiger oxidation is known to proceed with stereo-retention. Therefore, the process described in this paper is simply a combination and logical extension of two fundamental, text-book level reactions.

Answer: It is correct that Diels–Alder reaction and Baeyer–Villiger oxidation are well-known reactions. These reactions have been available for many decades. They were, however, not used or reported in a combined manner to generate polypropionates before, and there lies our novelty in this work. We conceived a procedure that could produce the alternating functionalities found in polypropionates (up to four contiguous chiral centres) in a ring system and transform it further into a structure that could be readily opened to its ultimate linear form, all of it being performed in a convenient one-pot operation. It makes logical sense, that’s true, and yet it is entirely possible that no one else has considered or has the realization that it is feasible to make polypropionates in this way, unless the relay concept as provided in our paper is laid out to them. We would also add that the critical *exo*-selectivity in Diels–Alder reaction—known to traditionally provide *endo*-selectivity—is only becoming understood very recently. We understand and respect the opinion of the reviewer—these are simple, textbook-level reactions—but we disagree, an elegant combination of simple reactions can at times amount to something new.

2. No enantio-control: The process depends simple on relative stereochemical control, and absolute chirality control is not achieved. In the application to etnangien, racemic starting material 37 was used. Possible reason for the facial selectivity (diastereoselectivity) of the diene side is not described.

Answer: Our initial intention is mainly to demonstrate that our method could generate polypropionates of the desired relative stereochemistry. To this end, it does not really matter whether we use a racemic or an enantiomerically pure starting material. In light of the comment by the referee, we repeated the preparation of the etnangien fragment using (*S*)-3-phenylbutanoic acid as starting material through the Diels–Alder/Baeyer–Villiger relay process. Herein the right absolute configurations of the contiguous stereocentres were achieved, attesting to the robust utility of the method. We also extended the

transformation not only to the C4–C9 fragment of etnangien but up to the C1–C9 fragment containing six contiguous stereocentres. The stereochemical control of the Diels–Alder reaction was likely attained via asymmetric induction by the substituents of the chiral carbon attached to C1 of diene **39**. A proposed reason for the facial selectivity in the *exo*-Diels–Alder reaction, which allowed the formation of the major diastereomer, is now added as Supplementary Fig. 2 and mentioned in the main text.

3. Exaggerated introduction: In the introductory section, the authors describe that repeating the process from product **6** (or **14**) will provide more elaborated structures. However, this is not achieved. Although the authors imply that previous methods for polyketide synthesis are step-consuming and not of atom-economy, the reported process also required multiple steps with using not-atom economically friendly reagents. The authors mention that the desired stereoisomers can be synthesized at will, which is again not achieved in the paper.

Answer: We toned down the claims in the introductory portions specified by the reviewer.

The reiteration of the Diels–Alder/Baeyer–Villiger relay process is conceptually sound, in principle, and future experiments may test this possibility, but not in the present work.

Regarding the step and atom economy, our report did not challenge previous methods having these qualities; there is no claim in the paper that we aimed at step/atom economy or our work is better than those other methods. We mainly specified that our method offers a convenient entry into polypropionates; our method is operationally simple and can be done in one pot. It is desirable to have an array of alternative methods to achieve something similar because natural product-related molecules are notoriously diverse and a method may work on the synthesis of one but not on another. Finally, although the word combination “at will”, now removed, might be too forceful, we have in fact synthesized our desired stereoisomer for the etnangien fragment in the present paper.

REVIEWERS' COMMENTS:

Reviewer #1 (Remarks to the Author):

I have examined the revised manuscript and the point-by-point response. The asymmetric synthesis of etnangien fragment improved the quality of this paper significantly. I am convinced that this method can be successfully applied to a total synthesis of a complex natural product, if the authors would invest some more efforts and time. I am happy with the current shape of the paper for publication in Nat. Commun. I do not think the questions from other reviewers about X-ray analysis and "new" concept really make sense for a piece of synthesis work.

Reviewer #2 (Remarks to the Author):

The authors have addressed my comments from the previous round of refereeing in regard to the crystallographic discussion. The X-ray experimental is still a little light: the authors should at least add the standard table of refinement parameters for each structure as supplementary information. Besides this I have no other issue with the crystallography - all the datasets are fairly routine collections - and with some small corrections below I can't see any other issues to prevent publication.

I would just request a small correction in the main text and supplementary information pertaining to chirality - the crystal structure for "41" (as referred in the text, line 153), as refined as an inversion twin, is actually the structure of 41 + enantiomer, and since 41 is represented with absolute configuration defined here the discussion should be amended to avoid misleading the reader. Similarly for "42" (+ enantiomer?), although no twin law was needed in the refinement, the uncertainty on the Flack parameter is large (due to the use of Mo Ka radiation, where absolute configuration of light atom structures cannot really be determined), so this doesn't prove the absolute configuration either, though the authors correctly refer to this structure as showing the relative configuration only.

On the subject of chirality, a chiral HPLC trace of the final product 48 would be a useful piece of supporting information to establish the %ee of the entire procedure and support its utility in the synthesis of etnangien.

Reviewer #3 (Remarks to the Author):

This reviewer agrees that the submitted paper is a solid synthetic work, but cannot agree that it involves conceptually novel items. This reviewer has recognized that Nature Communications are one of the top journals in organic synthesis, but the paper deals with 1990's chemistry. There is a fundamental disagreement between the authors and this reviewer about the definition of "conceptual novelty".

Response to Referees

Reviewer 1

Comment: I have examined the revised manuscript and the point-by-point response. The asymmetric synthesis of etnangien fragment improved the quality of this paper significantly. I am convinced that this method can be successfully applied to a total synthesis of a complex natural product, if the authors would invest some more efforts and time. I am happy with the current shape of the paper for publication in Nat. Commun. I do not think the questions from other reviewers about X-ray analysis and "new" concept really make sense for a piece of synthesis work.

Answer: We deeply appreciate the helpful comments by the reviewer, which allowed us to improve our paper.

Reviewer 2

Comment 1: The authors have addressed my comments from the previous round of refereeing in regard to the crystallographic discussion. The X-ray experimental is still a little light: the authors should at least add the standard table of refinement parameters for each structure as supplementary information. Besides this I have no other issue with the crystallography - all the datasets are fairly routine collections - and with some small corrections below I can't see any other issues to prevent publication.

Answer: We thank the reviewer for the helpful comments. The structure refinement parameters for each X-ray structure are now included as Supplementary Tables in the Supplementary Information. All of these Supplementary Tables are referenced in the main text.

Comment 2: I would just request a small correction in the main text and supplementary information pertaining to chirality - the crystal structure for "**41**" (as referred in the text, line 153), as refined as an inversion twin, is actually the structure of **41** + enantiomer, and since **41** is represented with absolute configuration defined here the discussion should be amended to avoid misleading the reader. Similarly for "**42**" (+ enantiomer?), although no twin law was needed in the refinement, the uncertainty on the Flack parameter is large (due to the use of Mo K α radiation, where absolute configuration of light atom structures cannot really be determined), so this doesn't prove the absolute configuration either, though the authors correctly refer to this structure as showing the relative configuration only.

Answer: The X-ray structure for compound **41** was obtained from the product of the synthetic sequence that started from the enantiomerically pure (*S*)-3-phenylbutanoic acid. As far as we could tell, there is no plausible way that the enantiomer of **41** would form along the synthetic procedure that we implemented in this paper. The original refinement for the crystal of **41** as an inversion twin was a mistake. Hence, the crystal data was refined again to reflect this fact, and the updated files were re-submitted to CCDC. The X-ray structure in Figure 4b was also updated. Nevertheless, as the reviewer pointed out, our use of Mo K α radiation prevented the full determination of absolute configuration. Thus, we modified the manuscript text to say that X-ray data mainly supported the structure and the relative configuration of the contiguous stereocentres of the molecule. We also changed the text in the Method section concerning the X-ray characterization of compound **41**. The same changes in the main text and the Supplementary Methods were carried out with respect to compound **42**.

Comment 3: On the subject of chirality, a chiral HPLC trace of the final product **48** would be a useful piece of supporting information to establish the %ee of the entire procedure and support its utility in the synthesis of etnangien.

Answer: Because we started our synthesis of the C1–C9 fragment of etnangien using the enantiomerically pure (*S*)-3-phenylbutanoic acid, we find no plausible way that the enantiomer of **48** could form along our synthetic procedure. We, therefore, believe that a chiral HPLC analysis to determine %ee is unnecessary. For this analytical procedure to succeed in our view, we would need to completely re-synthesize **48** using a racemic starting material to obtain the racemic product (we have not done this yet), which we will need for the optimization of the chiral chromatographic separation and later as an analytical standard. Such an endeavor will take quite a while to perform and is an undue burden just to prove enantiometric purity in a synthesis that is highly unlikely to generate an enantiomeric mixture.

Reviewer 3

Comment: This reviewer agrees that the submitted paper is a solid synthetic work, but cannot agree that it involves conceptually novel items. This reviewer has recognized that Nature Communications are one of the top journals in organic synthesis, but the paper deals with 1990's chemistry. There is a fundamental disagreement between the authors and this reviewer about the definition of "conceptual novelty".

Answer: The helpful comments of the reviewer are truly appreciated. We respect his/her opinion about the conceptual novelty of our work.